 

# Unexplored viral diversity in Siberian cranes and wild geese: metagenomic insights from a global wintering haven

Jing Gao,[1] Weijie Han,[2] Xiaojie Jiang,[3] Yuan Xi,[3] Yue Chen,[1] Shiyin Huang,[1] Xiaofeng Huang,[2] Yang Zhang,[2] Tianxiang Zhang,[2] Manyu Zhang,[2] Wen Zhang,[1] Bin Ni[1]

**ABSTRACT** Migratory birds are critical global carriers and vectors of various viruses, potentially driving the mutation and transmission of novel pathogens, especially zoonotic viruses. Despite advancements in sequencing technologies, the viral diversity in wintering Siberian cranes (*Leucogeranus leucogeranus*) and wild geese (*Anser* spp.) remains poorly understood. In this study, we conducted a viral metagenomic survey of fecal samples from 320 Siberian cranes and wild geese wintering in Poyang Lake, China. Through this approach, we identified 183 novel viruses associated with known and putative vertebrate-infecting viruses, including a novel coronavirus, parvoviruses, picornaviruses, picobirnaviruses, anelloviruses, and CRESS-DNA viruses. Furthermore, we detected evidence of cross-species transmission and identified viruses with zoonotic potential, such as picobirnaviruses and picornaviruses. These findings highlight the significant public health risks posed by migratory birds and provide new insights into the viral diversity within these populations, contributing to a better understanding of their role in viral evolution and transmission.

**IMPORTANCE** Understanding the diversity of enteroviruses in Siberian cranes and geese is essential for biodiversity conservation and ecosystem stability. As migratory birds, these species play key roles in ecological networks while carrying intestinal viruses that may spread along migration routes, which could pose potential risks to wildlife, poultry, and human health. This study systematically analyzed enterovirus diversity and the genetic characteristics of novel viruses in wintering Siberian cranes and geese at Poyang Lake using metagenomic sequencing. We identified viral sequences distantly related to known viruses and those with potential cross-species transmission risks. These findings highlight the diversity of migratory bird viruses and their public health implications, providing data to evaluate transmission risks and monitor emerging threats, supporting strategies for wildlife conservation and disease prevention.

**KEYWORDS** viral metagenomics, Siberian cranes, diversity, phylogenetic analysis, major global wintering site

According to various scientific models and estimates, the total number of viruses on Earth is estimated to be around $10^{31}$, making them among the most abundant and diverse biological entities on the planet (1, 2). However, research into viral diversity is still in its early stages, with less than 1% of potential viral species identified so far (3). Viruses play roles as both regulators and participants in ecosystems and have a profound impact on host populations, species interactions, and biogeochemical cycles (4, 5). At the same time, the high variability of viruses allows them to generate new variants through mutation or genetic recombination, which can lead to sudden public health crises as well as affect the stability of the global economy and ecosystems (6, 7). The global pandemic of SARS-CoV-2 and recurrent outbreaks of highly pathogenic avian influenza

**Peer Reviewer** Walter Harrington, St. Jude Children's Research Hospital, Memphis, Tennessee, USA

Address correspondence to Bin Ni, nibin@ujs.edu.cn.

Jing Gao, Weijie Han, and Xiaojie Jiang contributed equally to this article. The author order was determined by drawing straws.

The authors declare no conflict of interest.

See the funding table on p. 20.

viruses highlight a critical subset of RNA viruses—those with broad host ranges and high mutation rates—that exhibit rapid spread and an ability to exploit ecological interfaces created by anthropogenic disturbances (8, 9). These risks are amplified in multi-host ecosystems, where human activities such as wetland conversion, agricultural intensification, and pollution fragment natural habitats and force overlapping host ranges, thereby enabling viral persistence and recombination across taxa (10). Within such perturbed ecosystems, pathogens increasingly traverse species boundaries via environmental reservoirs (e.g., contaminated water and bioaerosols), a dynamic exemplified by the co-detection of novel avian-associated viruses—including coronaviruses and picornaviruses—alongside known zoonotic threats (11). These pathways contribute significantly to the emergence and spread of zoonotic diseases (12). Therefore, in-depth studies on viral diversity, their roles in ecosystems, and potential transmission pathways are needed to better predict and mitigate future viral threats.

Migratory birds are important hosts and vectors for virus transmission, and their migratory behavior and habitat use characteristics play a key role in the trans-regional spread and mutation of viruses (13, 14). Avian influenza viruses (AIVs) have historically garnered attention due to their potential zoonotic and economic ramifications, exemplified by the 2005 H5N1 outbreak at Qinghai Lake, which resulted in significant mortality among wild birds and subsequently disseminated across continents (15, 16). Migratory birds undertake long-distance migrations across multiple ecosystems, coming into contact with water, soil, and other animals, providing ideal conditions for virus transmission and recombination (13). Some pathogens can be transmitted to birds through contaminated water and waste, where they colonize the intestinal tract (17). These pathogens can then spread to the farm environment through the birds' excreta, which may infect poultry and humans. In addition to avian influenza, the Usutu virus and the West Nile virus also pose serious risks for zoonotic diseases (18, 19). Therefore, systematic surveillance of viral infections in birds is crucial for preventing potential outbreaks and enabling rapid responses to epidemics.

As one of the largest wintering sites along the East Asia-Australasia migratory route, Poyang Lake attracts over 600,000 migratory birds annually (20). Siberian cranes are currently listed as critically endangered by the International Union for Conservation of Nature (IUCN) (21). More than 98% of the eastern Siberian cranes winter at Poyang Lake (22). Several species of wild geese also share the habitat with Siberian cranes. Together, they form the primary migratory bird population at Poyang Lake. Due to their high group density and frequent interactions with other birds and poultry, these species are potential vectors for virus transmission, particularly during the wintering period (23, 24). During this time, Poyang Lake experiences a dry season, causing the water surface area to shrink, leaving behind herbaceous meadows, mudflats, and isolated water bodies (23). This reduction in water surface area significantly facilitates the spread of pathogens (25). Several viral families, including adenoviruses, circoviruses, and coronaviruses, have been detected in environmental samples from Poyang Lake, indicating the area's rich viral diversity and highlighting potential public health risks (20). However, research on the viral communities of these endangered migratory birds is limited, particularly regarding the systematic study of the diversity of Siberian cranes and goose enteroviruses and their genetic characterization. Recent advances in viral metagenomics offer a powerful approach to uncover viral diversity. Metagenomics reveals previously unknown viral species and their ecological roles by analyzing genetic material directly from environmental or host samples (26). This technique has expanded our understanding of virus-host interactions and the diversity of viral populations, including the human virome, gut microbiome, and environmental phages (26, 27). Applying metagenomics to study the diversity and genetic characteristics of enteric viruses in Siberian cranes and geese could provide valuable insights.

This study focuses on the Siberian cranes and wild geese at Poyang Lake, using viral metagenomics to analyze their gut virus communities. It aims to uncover viral diversity, explore potential cross-species transmission, and examine the link between

viral dynamics and migratory bird ecology. The findings will enhance understanding of migratory bird virus ecosystems, inform global health surveillance, and aid in controlling emerging infectious diseases and conserving bird habitats.

## MATERIALS AND METHODS

### Birds sample collection

Fecal samples were collected in collaboration with the Jiangxi Academy of Forestry from January to February 2024 at two major wintering sites in the Poyang Lake region: Baihezhou (28.8446°N–28.8500°N, 116.4671°E–116.4700°E) and the Siberian Crane Conservation Area at Wuxing Farm (28.7612°N–28.7650°N, 116.3290°E–116.3350°E). A total of 208 fecal samples were collected from Siberian cranes and 112 from geese, with detailed collection information provided in Table 1. Sampling locations were systematically selected to ensure even coverage of the target areas, with sampling efforts proportionate to the observed bird population density. Samples were collected non-invasively with binoculars to observe individual birds defecating and collect freshly deposited feces. The locations and times of collection were carefully recorded. Only feces associated with observed defecating birds were collected to minimize the risk of contamination or misidentification. Species identification was based on detailed field observations of bird morphology and habitat-specific ecological characteristics, such as feeding behaviors and flock composition. Experienced ornithologists, for accurate species identification, conducted observations. Continuous field monitoring and official health reports from the Jiangxi Academy of Forestry confirmed the absence of mortality or disease. Collected samples were immediately placed into sterile containers, temporarily stored at 4°C in portable refrigerators, and transported on dry ice to the laboratory within 8 hours. Upon arrival, samples were immediately stored at −80°C for viral metagenomic analysis, with all steps monitored to maintain cold-chain integrity.

### Sample preparation, library construction, and quality control

The 320 samples were pooled into 32 groups based on species classification. Each group contained an average of 10 samples. Each group was homogenized and subjected to three freeze-thaw cycles to promote cell lysis and viral release, followed by thawing on dry ice, and then, 100 mg of each sample was resuspended in 1 mL of Dulbecco's phosphate-buffered saline (DPBS). The supernatant was collected after centrifugation (10 min, 15,000 × g, 4°C) (28). To remove eukaryotic and bacterial cell-sized particles, the pools were centrifuged (5 min, 12,000 × g, 4°C), and the resulting supernatant was passed through a 0.45 µm filter (29, 30). This filtered supernatant was incubated with Turbo DNase (Thermo Fisher Scientific, MA, USA), Baseline-ZERO DNase (Epicenter, WI, USA), Benzonase Nuclease (Novagen, MA, USA), and RNase A (Thermo Fisher Scientific) at 37 °C for 1 hour to digest unprotected host and bacterial nucleic acids, enriching for encapsidated viral genomes prior to extraction (31–33). The remaining nucleic acids, including both DNA and RNA, were extracted using the QIAamp Viral RNA Mini Kit (Qiagen), which allows for the co-purification of viral RNA and small DNA genomes, following the manufacturer's instructions. RNA was converted to complementary DNA (cDNA) using the Superscript IV Reverse Transcriptase Kit, which utilizes random hexamers as primers for first-strand synthesis. Double-stranded DNA (dsDNA) was then synthesized from the cDNA to construct the DNA library. For single-stranded

**TABLE 1** Sample collection information

| Sampling site | Species | Sample no. | Proportion (%) |
| --- | --- | --- | --- |
| Baihezhou | Siberian crane | 41 | 36.6 |
| | Goose | 71 | 63.4 |
| Wuxing Farm | Siberian crane | 167 | 80.3 |
| | Goose | 41 | 19.7 |

DNA (ssDNA) viruses, dsDNA was generated using the Klenow polymerase reaction (New England Biolabs). The Nextera XT DNA Sample Preparation Kit (Illumina) prepared the sequencing libraries by pooling 32 dsDNA products. These pooled libraries were sequenced on an Illumina NovaSeq 6000 platform using 150 bp paired-end reads, with each pool uniquely identified by dual barcodes. Stringent precautions were taken throughout the experimental process to prevent cross-contamination and nucleic acid degradation. Aerosol-resistant filter tips were employed to minimize the risk of sample contamination, and all materials that directly contacted nucleic acid samples, such as microcentrifuge tubes and pipette tips, were certified DNase- and RNase-free. The samples were dissolved in diethylpyrocarbonate (DEPC)-treated water supplemented with RNase inhibitors (28). Sterile ddH$_2$O was prepared in parallel and processed identically under the same experimental conditions as blank controls. Quality assessments were conducted using agarose gel electrophoresis and the Agilent Bioanalyzer 2100, with no detectable DNA in the control group. During sequencing on the Illumina MiSeq or HiSeq platform, the control group produced an extremely low number of reads. Further BLASTx analysis confirmed the absence of viral sequences in the control pool.

## Bioinformatics analysis

Data were processed through an internal analysis pipeline on a 32-node Linux cluster. First, the raw sequencing data from the 32 libraries were quality assessed using FastQC (v0.11.9) (34) and integrated to generate quality reports using MultiQC (v1.11) (35). Sequences were then subjected to splice removal and low-quality region trimming using Trim Galore (v0.6.5) (https://www.bioinformatics.babraham.ac.uk/projects/trim_galore/), with trimming parameters set to "--phred33," "--length 35," "--stringency 3," "--fastqc." and "--paired." To eliminate host-derived contamination, reads were aligned to a comprehensive reference database of eukaryotic and prokaryotic genomes using Bowtie2 (v2.3.4.1) (36). In this study, the term "host" refers broadly to all non-viral genomic material present in fecal samples, including sequences from avian hosts and associated microbial flora. Only non-host reads were retained for downstream virome analysis. *De novo* assembly was performed using MEGAHIT (v1.2.9) (37) with a minimum contig length of 200 bp. After that, the contigs and single reads were aligned to the NCBI viral protein database using DIAMOND BLASTx (v0.9.24) with the an E-value threshold set to of 1e$^{-5}$ and further screened by comparison to the non-viral, non-redundant protein databases to remove false positives (38). Unclassifiable sequences were screened against the vFam database using HMMER v3.1b2 to identify possible false negatives (39, 40). Finally, annotation information for virus-related contigs and singlet reads was presented using MEGAN (v6.22.2).

## Viral sequences extension and annotation

The previously obtained reads were classified into their respective taxonomic groups using MEGAN (v6.22.2). *De novo* assembly and reference mapping were subsequently performed with Geneious Prime 2024.0.7 (https://www.geneious.com), where individual contigs served as references to map the original data, thereby facilitating the assembly of reads into complete or partial viral genomes. The Find ORFs function in Geneious was employed to predict open reading frames (ORFs) in the viral sequences (minimum length: 300 bp; genetic code: standard; start codon: ATG). The predicted ORFs were then compared to the NCBI nr database using BLASTx. ORF annotations were generated based on comparisons with conserved domain databases using RPS-BLAST, with an *E*-value threshold set to <10$^{-5}$. Contigs annotated with viral marker genes from major viral taxonomic groups were selected, and the complete ORFs identified were used for subsequent phylogenetic analyses.

## Viral community analysis

Statistical analyses related to the experiments were performed using MEGAN v6.22.2 and R v4.4.1. Megan normalized and compared the compositional analyses of the 32

libraries (41). Alpha diversity and beta diversity analyses were performed using the vegan package, with statistical significance set at $P < 0.05$. Shannon indices were analyzed using the Wilcoxon test. Principal coordinate analysis (PCoA) based on Bray-Curtis phase dissimilarity was performed using Permute, Llattice, vegan, and ape packages. Results of viral community structure and abundance were visualized using heat maps, Venn diagrams, and histograms. These were generated using the pheatmap, Venn, and ggplot2 packages. Viruses shown as "shared" in Venn diagrams were defined as either ICTV-assigned species found in both hosts or novel viruses sharing ≥95% amino acid identity in conserved proteins (e.g., RdRp, Rep, NS1, and ORF1), based on BLASTp results ($E$-value $< 1e^{-10}$, coverage $> 80\%$).

## Phylogenetic analysis

Phylogenetic analyses were performed based on the predicted protein sequences of the viruses identified in this study and protein sequences of reference strains belonging to different virus groups downloaded from the NCBI GenBank database, as protein-level comparisons are more robust for highly divergent viral taxa. Relevant protein sequences were compared using MUSCLE in MEGA v.11.0.13 with default settings (42). MrBayes v.3.2.7 was then used to construct Bayesian inference trees (43). We used two parallel runs of Markov chain Monte Carlo (MCMC) sampling in MrBayes and set 'prset aamodelpr = mixed' for the phylogenetic analysis based on protein sequences. The run was terminated when the standard deviation of the split frequency was less than 0.01, and the top 25% of trees were discarded (44). In addition, maximum likelihood trees were constructed to support all Bayesian inference trees in the MEGA software. Phylogenetic trees were displayed using FigTree v.1.4.4 (http://tree.bio.ed.ac.uk/software/figtree/), Adobe Illustrator 2022 v.27.0, and iTOL v.6.

## Receptor-binding domain (RBD) structural analysis

The RBD amino acid sequences from CoronaCrane85, chicken infectious bronchitis virus (IBV), duck coronavirus, Yunnan coronavirus 2, and mute swan gammacoronavirus were aligned using MUSCLE. The RBD structure of the spike protein was predicted using ProMod3 on the SWISS-MODEL server ([https://swissmodel.expasy.org/](https://swissmodel.expasy.org/)), based on target-template alignments. Model quality was assessed using the QMEAN scoring function, which evaluates both global and per-residue accuracy. The predicted protein structures were visualized as PDB files and compared with those of other avian coronaviruses.

## Pairwise sequence identity analysis

To assess the nucleotide sequence similarity of the newly identified anelloviruses, pairwise identity analysis was performed. The viral sequences acquired in this investigation were aligned with highly similar sequences found via NCBI BLASTx with representative anelloviruses reference sequences retrieved from GenBank. Multiple sequence alignment was conducted using MUSCLE implemented in MEGA11. The aligned nucleotide sequences were subsequently analyzed using the Sequence Demarcation Tool (SDT v1.3) to calculate pairwise identity values and visualize sequence identity matrices.

## RESULTS

### Overview of the virome

A large-scale viral metagenomic survey was performed using 212 fecal samples from Siberian cranes and 108 from geese, all collected near Poyang Lake, one of the major global wintering sites for Siberian cranes. Thirty-two libraries were constructed from these samples and sequenced using the Illumina NovaSeq platform, generating 295,525,785 raw reads with an average GC content of 49.2%. The assembled

metagenomic data were analyzed using BLASTx ($E$-value $<10^{-5}$) to compare against the GenBank non-redundant protein database, identifying 10,815,724 viral reads, which accounted for 3.660% of the total reads. Species richness was assessed through rarefaction and accumulation curve analyses, which indicated that the observed viral species in most of the 32 libraries had reached a saturation point. This finding suggests that the current sequencing depth sufficiently covered all viral species in the collected samples, and additional sequencing data would not significantly increase the diversity of identified viral species (Fig. 1A). Furthermore, the species accumulation curve gradually plateaued with increasing sample size, indicating that the number of samples collected in this study was adequate and representative of the studied ecosystem (Fig. 1B). The accumulation curve also revealed the presence of over 1,500 distinct viral species across the 32 libraries.

Taxonomic analysis of the *de novo* assembled 63,816 viral contigs identified 172 viral genome sequences associated with known and putative vertebrate-infecting viruses, classified into 11 families: *Anelloviridae* ($n = 11$), *Circoviridae* ($n = 9$), Genomoviridae ($n = 11$), Parvoviridae ($n = 36$), Smacoviridae ($n = 3$), unclassified circular Rep-encoding single-stranded DNA (CRESS-DNA) viruses ($n = 27$), Coronaviridae ($n = 1$), *Astroviridae* ($n = 10$), *Caliciviridae* ($n = 13$), *Picobirnaviridae* ($n = 21$), and *Picornaviridae* ($n = 30$).

## Comparative analysis of viral communities

To investigate the distribution and abundance of viral families in Siberian cranes and wild geese, a heatmap was constructed based on the genomic sequences of viruses from 32 libraries, categorized at the family level and stratified by bird species and nucleic acid type. The analysis identified 100 viral families, including 36 dsDNA viral families, 13 ssDNA viral families, 34 ssRNA (+) viral families, four ssRNA (−) viral families, two ssRNA (RT) viral families, two dsDNA (RT) viral families, and nine dsRNA viral families (Fig. 2A). Comparisons between the viral communities associated with Siberian cranes and wild geese revealed notable differences in composition. Among the 32 libraries, 21 were derived from Siberian cranes and 11 from wild geese. The *Siphoviridae* was the most abundant in Siberian cranes, while Picornaviridae dominated wild geese. Additionally, Leviviridae and *Fiersviridae* were predominant in Siberian cranes, whereas *Siphoviridae* and *Microviridae* were more abundant in geese. The relative abundances of *Parvoviridae* and *Coronaviridae* were significantly higher in wild geese compared to Siberian cranes, which may reflect host-specific viral associations or ecological differences in viral exposure between the two species. These findings highlight distinct viral community structures between the two bird species despite some overlap in viral family composition (Fig. 2B). At the species level, a Venn diagram was used to compare the shared and unique viral species between the two bird hosts. A total of 1,630 viral species were identified, of which 550 were shared, accounting for 37.36% of the viral species in Siberian cranes and 77.68% in wild geese (Fig. 2C).

To explore the regional variations in the intestinal viral communities of Siberian cranes and wild geese, we conducted analyses of α-diversity and β-diversity. The gut viral diversity (Shannon index) of Siberian cranes was significantly higher than that of wild geese (Wilcoxon test, $P = 0.022$), indicating significantly greater species richness and evenness in the viral communities of Siberian cranes (Fig. 3A). Principal coordinate analysis (PCoA) based on Bray-Curtis dissimilarities revealed a significant separation in viral community structure between the two bird groups (PERMANOVA, $R = 0.325$, $P = 0.001$), likely driven by differences in host ecology (e.g., diet, migration, and habitat use) and exposure to viral sources (Fig. 3B).

## Unveiling the diversity of vertebrate-associated viruses

### Anelloviridae

Anelloviruses are small, single-stranded, circular DNA viruses with high genetic diversity, with ORF1/VP1 being the most conserved protein encoded by them (45). Although

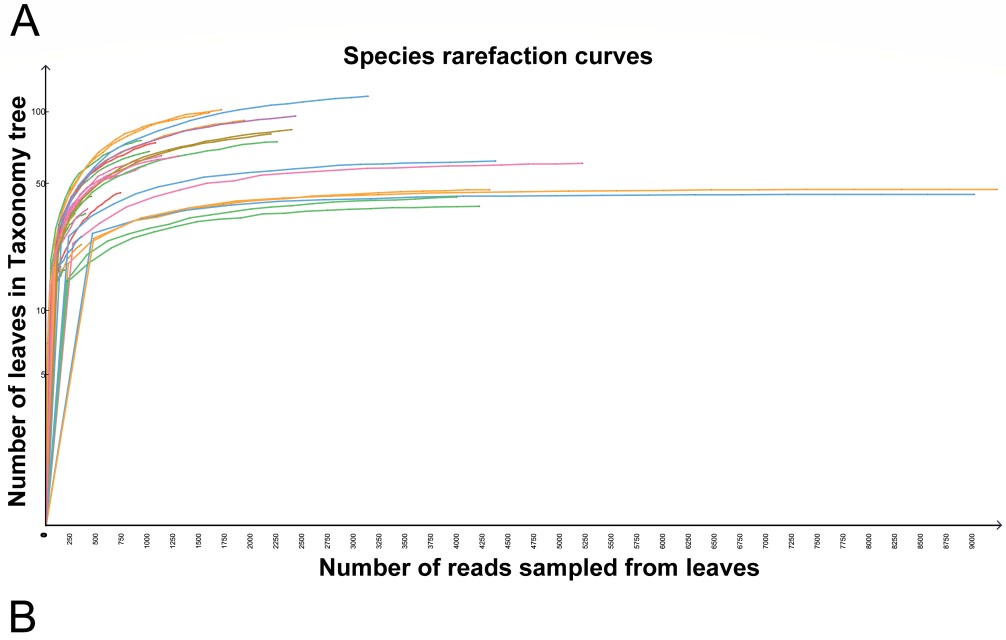

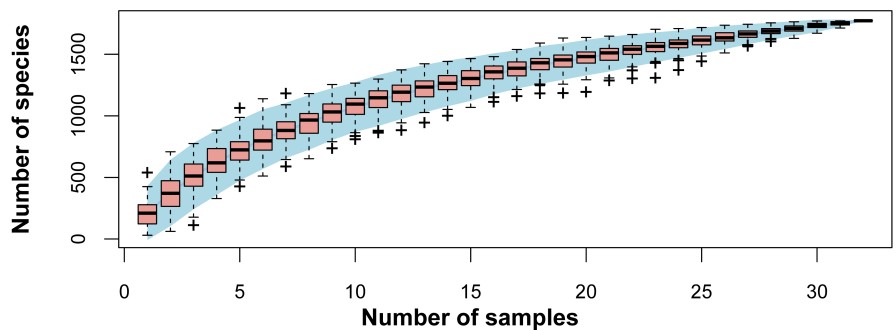

**FIG 1** Viral species diversity in 32 libraries. (A) Plot of the resulting species sparsity curves after logarithmic scale transformation of the raw data in Megan v6.21.16 software. (B) Accumulation curves of viral species in avian macrogenomes. Individual box-and-line plots correspond to the richness values of the samples, and the light blue areas represent 95% confidence intervals.

no definitive evidence links anelloviruses directly to specific diseases, their complex interactions with the host immune system suggest that this viral family may have potential pathogenicity (46). Through metagenomic analysis, we obtained the genomes of nine anelloviruses from wild geese and two from Siberian cranes. We then constructed a phylogenetic tree based on the amino acid sequence of VP1 (Fig. 4) . The phylogenetic tree reveals that these anelloviruses cluster with gyroviruses and are divided into three evolutionary clades (Fig. 4). In particular, AnelloCrane74 shows evolutionary relatedness to a bat gyrovirus genome in the phylogenetic tree. The genus *Gyrovirus* was classified within the Anelloviridae family in 2017 (47). They can infect various poultry species, including chickens, ducks, and turkeys, as well as certain wild bird species, typically transmitted via the fecal-oral route (48). Currently, the only known member of the *Gyrovirus* is the chicken anemia virus (CAV) (49). Further research in virology may lead to the discovery of more novel gyrovirus species. A pairwise comparison based on the VP1 nucleotide sequences of the newly identified gyrovirus with known gyrovirus reveals an identity below 60% (Fig. S1). According to the recently proposed species demarcation threshold for gyroviruses (69% nucleotide identity for VP1), these newly identified

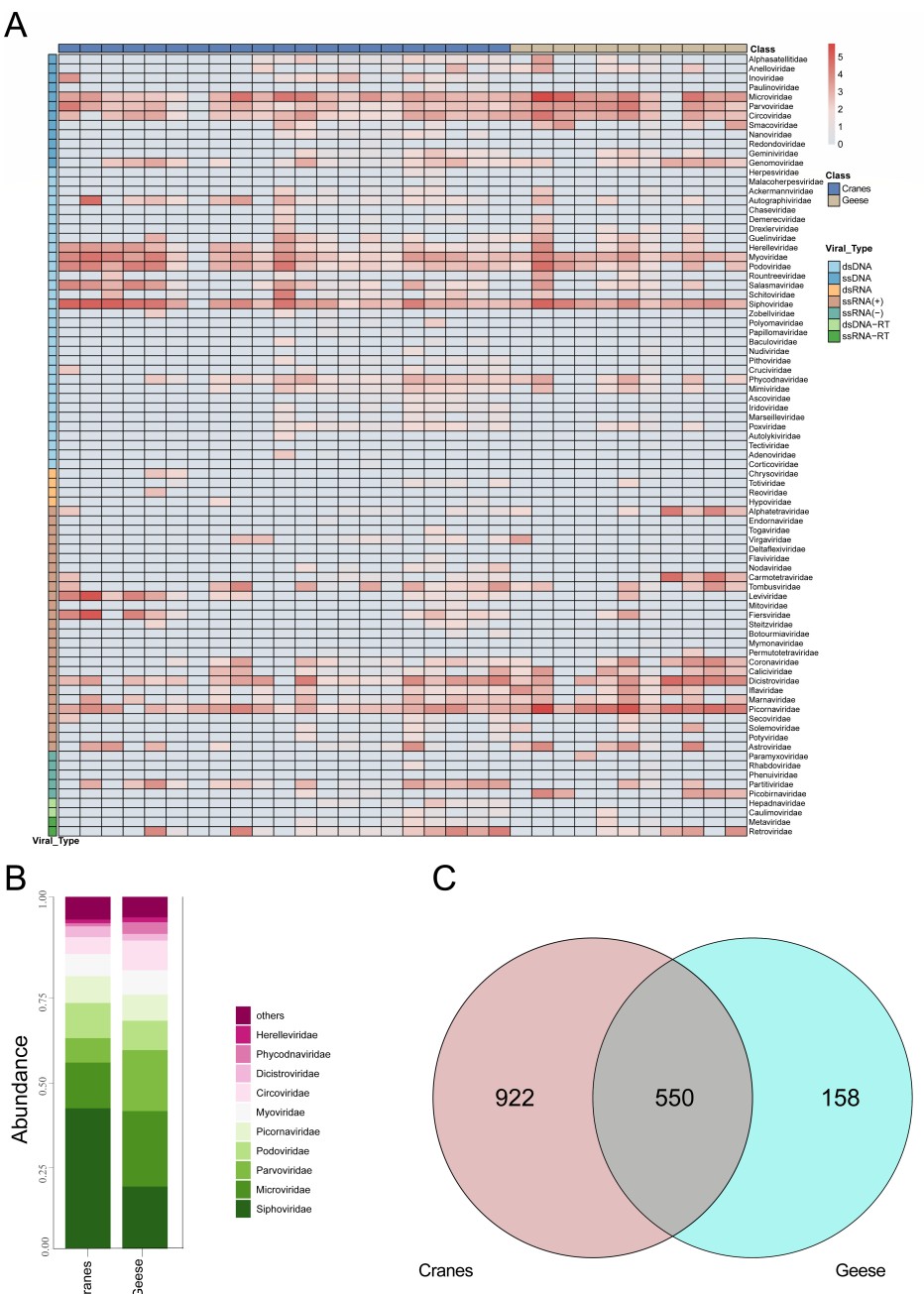

**FIG 2** Classification of viral reads at the family or species level. (A) Heatmap constructed using the log10-transformed read counts of each viral family in individual libraries. Annotations for nucleic acid types, viral families, and bird species are color-coded (refer to the legend for details). (B) Bar chart illustrating the relative proportions and classifications of viral families, categorized by bird species. (C) Venn diagram showing distribution of shared and distinct viral sequences.

anelloviruses represent novel species within the genus (49). Notably, six of these newly characterized gyroviruses were found to be monophyletic, with high statistical support.

## *Astroviridae*

The *Astroviridae* contains two genera: *Mamastrovirus*, which infects mammals, and *Avastrovirus*, which infects birds. They primarily cause gastrointestinal disease (50). Migratory birds, as critical vectors, facilitate interspecies transmission by shedding the astroviruses in feces along their migratory routes, potentially impacting wild birds,

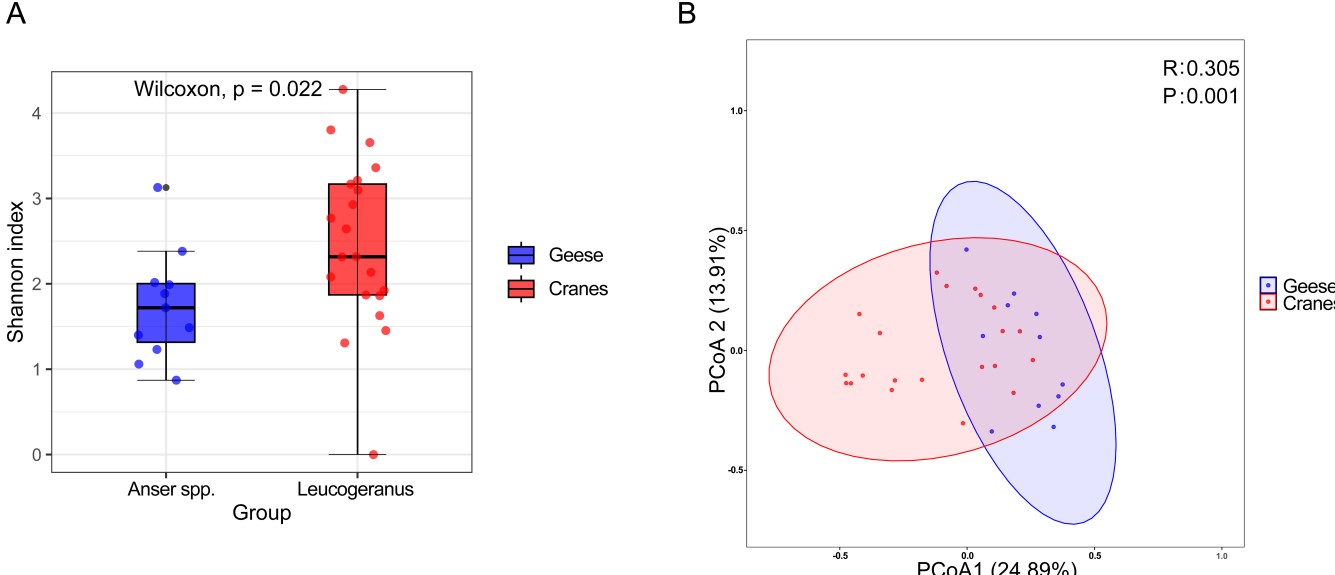

**FIG 3** Diversity of viral communities between Siberian cranes and wild geese. (A) Before comparing viral α-diversity, viruses to be compared were standardized using Megan, and viral abundance (genus level) was measured using Shannon's index for Siberian cranes and wild geese, which were divided into different bird species. *P*-values were calculated using the Wilcoxon test. (B) Before comparing viral β-diversity, the viruses to be compared were standardized using Megan's and principal coordinate analysis (PCoA) was performed at the family level for Siberian cranes and wild geese. *r* greater than 0 indicates a difference between the groups. The study was considered statistically significant if the *P* value was less than 0.05.

poultry, and humans, posing zoonotic risks (51). We identified and characterized eight astroviruses from Siberian cranes and wild geese, including four with complete genomes. Phylogenetic analysis of the RdRp region revealed that these astroviruses share evolutionary origins with astroviruses previously identified in avian hosts, such as chickens, ducks, and swans. Notably, AstroCrane81 exhibited 81.52% amino acid identity to a swan astrovirus genome collected in the United Kingdom (GenBank no. MW588064), and AstroGoose93 clustered closely with an astrovirus identified in river water in New Zealand (GenBank no. OM954094), sharing a common branch in the phylogenetic tree (Fig. 5A). AstroGoose95V2 forms a separate branch in the evolutionary tree. In the evolutionary tree of the capsid, AstroGoose94 and AstroGoose87 formed separate evolutionary branches, respectively. AstroCrane67 and AstroCrane81 clustered together and shared less than 50% amino acid sequence identity with swan viruses collected from mute swans (Fig. 5B). The topologies of the phylogenetic trees constructed from the Cap and RdRp proteins were very similar, indicating that no recombination has occurred in these astroviruses. According to ICTV, the taxonomy of Avastrovirus species is under revision. Genetic analyses of complete capsid regions indicate that avian astroviruses can be classified into two major genotypes (Genotype I and Genotype II), with an average pairwise amino acid genetic distance (p-distance) of 0.704 ± 0.013 (17). For the four complete genomes identified in this study, phylogenetic analyses of the capsid protein revealed genetic distances ranging from 0.161 to 0.600, confirming they fall within known genotypic diversity without forming novel clades (Table S1).

## *Coronaviridae*

Coronaviruses are a group of positive-sense RNA viruses that infect a wide range of animals and humans, causing respiratory, gastrointestinal, and nervous system diseases (52). Three major coronavirus outbreaks in humans have involved animal-to-human transmission, posing substantial public health risks (53–55). Coronaviruses are classified into four genera: *Alphacoronavirus*, *Betacoronavirus*, *Gammacoronavirus*, and *Deltacoronavirus* (52). Among them, Gammacoronaviruses primarily infect birds but can also infect

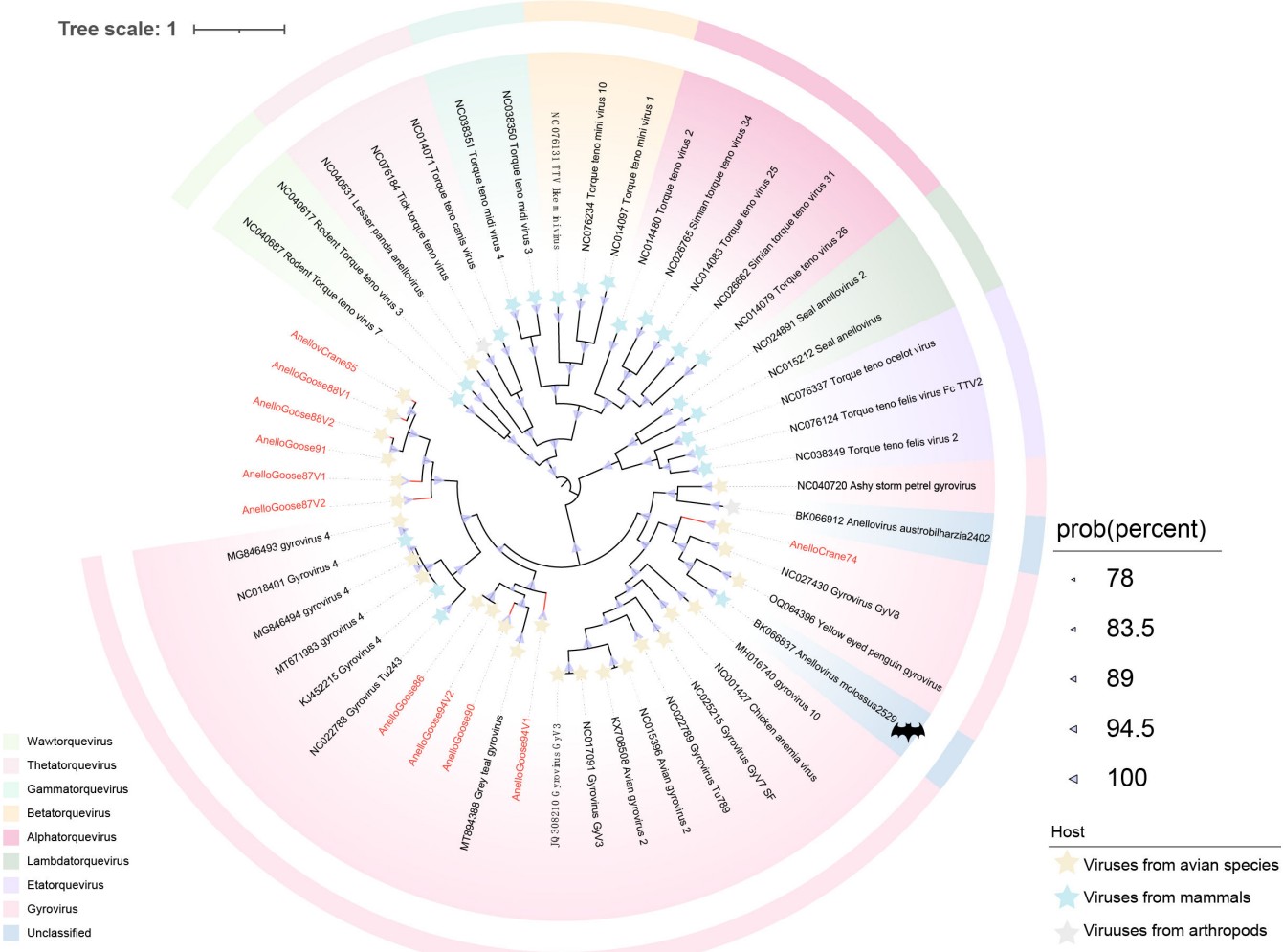

**FIG 4** Phylogenetic relationships of *Anelloviridae*. Bayesian inference tree constructed based on the VP1 amino acid sequences of *Anelloviridae*. Sequences identified in this study are highlighted in red. Relevant annotations are provided in the accompanying legend.

certain mammals and remain relatively understudied (56). In this study, a complete coronavirus genome, tentatively named CoronaCrane85, was identified in the feces of Siberian cranes. The genome is 28,981 nucleotides in length and encodes multiple nonstructural and structural proteins (Fig. 2; Fig. S2). It contains two major open reading frames, ORF1a and ORF1b, which encode nonstructural proteins involved in viral replication and transcription, such as RNA-dependent RNA polymerase (RdRp) and protease. The structural proteins include spike (S), membrane (M), envelope (E), and nucleocapsid (N) proteins, which are essential for viral entry, assembly, and stability (57). To explore the evolutionary relationships of Gammacorona1792, we constructed phylogenetic trees based on the amino acid sequences of ORF1a, ORF1b, and the major structural proteins (S, E, M, and N) (Fig. 6; Fig. S4). In all phylogenies, CoronaCrane85 clustered with members of the genus *Gammacoronavirus*, including members of the IBV, as well as other bird gammacoronaviruses. However, only in the S and 1b protein phylogenies did Gammacorona1792 cluster with a mute swan gammacoronavirus (GenBank no. MW588092) to form a distinct branch. The amino acid identities between CoronaCrane85 and the mute swan gammacoronavirus were 97.58% in the 1b protein and 89.31% in the S protein, indicating a higher degree of divergence in the spike gene.

To better understand the mutation sites and potential host specificity of Corona-Crane85, the receptor-binding domain (RBD) of its spike protein was compared with

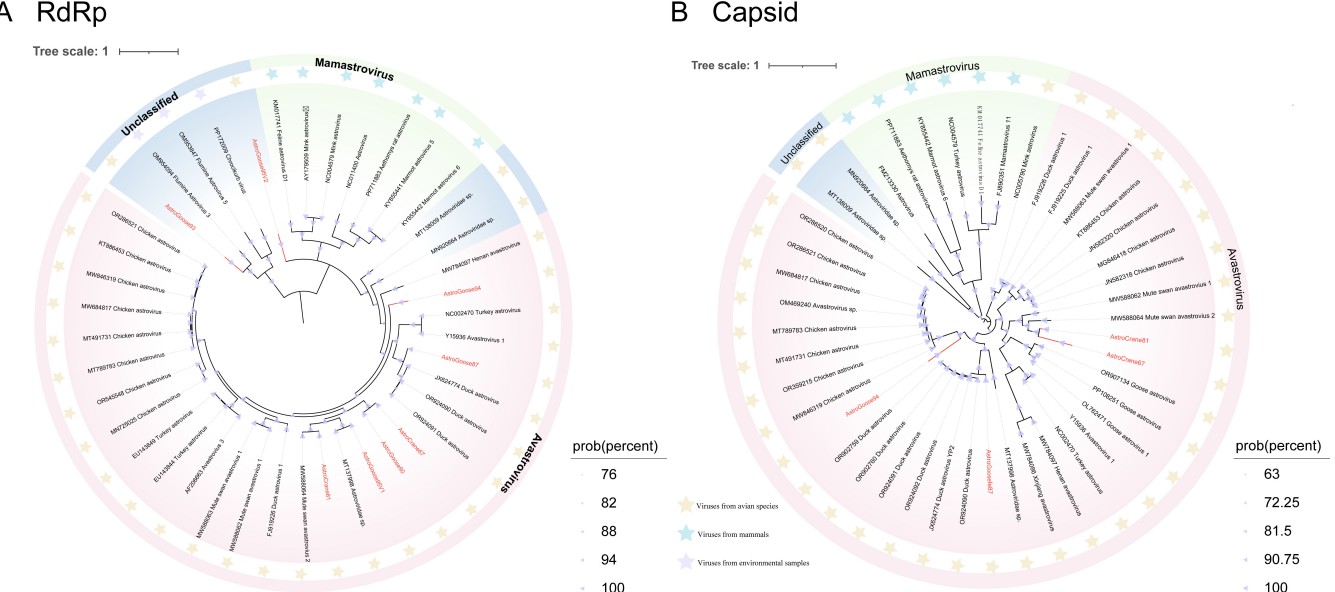

**FIG 5** Phylogenetic relationship of *Astroviridae*. (A) Bayesian inference tree based on the RdRp protein. (B) Bayesian inference tree based on the Capsid protein. The red color represents the sequences in this study. Relevant annotations are provided in the accompanying legend.

those of other avian coronaviruses (Fig. S3). The results indicated that CoronaCrane85 is more similar to the mute swan gammacoronavirus at the amino acid level. Both gammacoronaviruses exhibit amino acid deletions at positions 346–352, 373–379, and 458–462; insertions at positions 341–343 and 441–443; and mutations at several highly conserved sites, including Y472, G419, and V437. Notably, the V437I substitution was uniquely observed in Gammacorona1792. We used the SWISS-MODEL server to predict the three-dimensional structures of the spike protein RBDs of Gammacorona1792, Yunnan coronavirus 2, IBV, duck coronavirus, and mute swan gammacoronavirus via protein homology modeling. The predicted RBD structures of Gammacorona1792 and mute swan gammacoronavirus were highly similar, with a notable difference at theCaliciviruses are a class of non-enveloped, single-stranded, positive-stranded RNA viruses mutation site V473, which distinguishes Gammacorona1792 from the others.

## Caliciviridae

Caliciviruses are a class of non-enveloped, single-stranded, positive-stranded RNA viruses (58). This family includes several significant genera, such as *Norovirus*, *Sapovirus*, *Vesivirus*, and *Lagovirus*, which can infect various mammals and birds, often causing gastrointestinal diseases. Previous studies have demonstrated that *Caliciviridae* exhibits a strong potential for cross-host transmission, particularly among mammalian species (59, 60). In recent years, many novel unclassified caliciviruses have been widely detected in wild birds (61), geese (62), and fish (63). In this study, we characterized three caliciviruses (including two complete genomes) from Siberian cranes and ten (including five complete genomes) from wild geese. Based on the phylogenetic analysis of 13 strains with complete RdRp regions and nine strains with complete VP1 regions, these newly identified caliciviruses form a distinct lineage from the unclassified Caliciviridae (Fig. 7).

## Parvoviridae

Parvoviruses are single-stranded DNA (ssDNA) viruses with compact genomes ranging from 4 to 6 kb (64). They have a broad host range, including birds, mammals, reptiles, and invertebrates (65). Among humans, parvoviruses are represented by human parvovirus B19, which is known to cause erythema infectiosum (commonly referred to as the "fifth

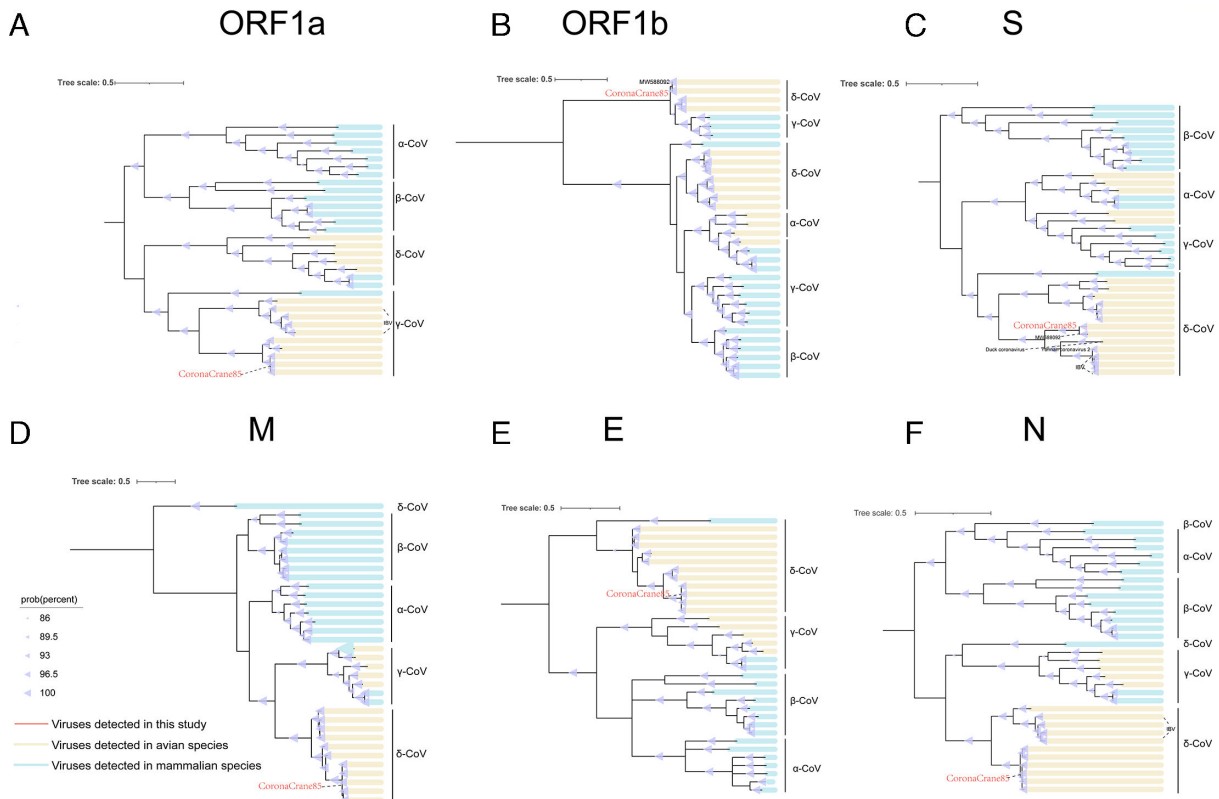

**FIG 6** Phylogenetic relationship of *Coronaviridae*. Maximum likelihood phylogenetic trees of protein of the ORF1a (A), ORF1b (B), S (C), M (D), E (E), and N (F) genes of CoronaCrane85 and related coronaviruses. Different-colored lines indicate the corresponding hosts, with further details provided in the legend located at the bottom right.

disease") (66). Among animals, parvoviruses are often associated with clinical symptoms, such as growth retardation and watery diarrhea, and they have been described in various avian species (67). Migratory birds are believed to act as significant vectors due to their long-distance travel and ability to carry pathogens, facilitating the dissemination of environmentally resistant viruses, such as parvoviruses, which can be transmitted through fecal shedding or direct contact (68). In this study, we used viral metagenomics to characterize 13 parvoviruses (including eight complete genomes) from Siberian cranes at Poyang Lake and 24 parvoviruses (including 17 complete genomes) from wild geese. To determine the evolutionary relationships of these viruses, we constructed a phylogenetic tree based on the amino acid sequences of the nonstructural protein 1 (NS1) domain (Fig. 8). The phylogenetic analysis revealed that the 36 newly identified parvoviruses were distributed across 11 genera, including three genera within the subfamily *Parvovirinae*: *Aveparvovirus* (*n* = 1), *Dependoparvovirus* (*n* = 1), *Parvovirinae* sp. (*n* = 1), and *Chapparvovirus* (*n* = 11). According to the ICTV classification criteria, parvoviruses are considered members of the same species if their NS1 protein sequences share more than 85% amino acid identity. Members of the same genus should share at least 35–40% amino acid identity in their NS1 sequences, with a coverage of over 80% (69). Among the parvoviruses identified in this study, only ParvoGoose91V10 exhibited an NS1 protein identity of 96.42% with an aveparvovirus (GenBank no. MW588065) previously detected in mute swans in Dorset, UK. The remaining 36 parvoviruses shared 29.14–80.73% identity with their closest database matches, suggesting they represent new species. Moreover, ParvoCrane74, ParvoCrane81V2, and ParvoeGoose94 formed distinct evolutionary branches in the phylogenetic tree, with NS1 sequence identities below 38%, indicating they may represent three new genera. Further phylogenetic analysis revealed evolutionary relationships between some of the identified parvoviruses

## A  RdRp

## B  VP1

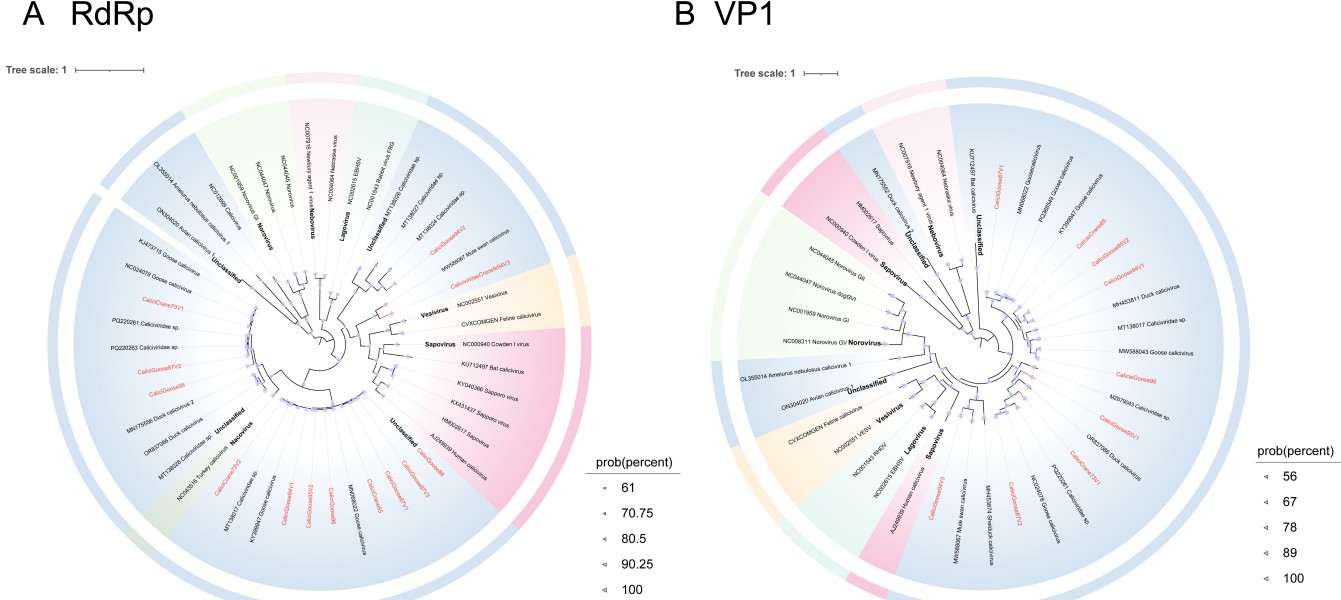

**FIG 7** Phylogenetic relationship of *Caliciviridae*. A Bayesian inference tree was constructed based on the amino acid sequences of the (A) RdRP and (B) VP1 from *Caliciviridae*. Sequences identified in this study are highlighted in red. Relevant annotations are provided in the accompanying legend.

and other vertebrate parvoviruses. For example, ParvoCrane65V2 shared 75.13% amino acid sequence similarity with a parvovirus (GenBank no. NC022089) from non-A-E hepatitis in Chongqing, China. ParvoGoose89V2, ParvoGoose89V3, and ParvoCrane65V3 were found to be evolutionarily related to canine protoparvovirus genomes. Additionally, ParvoGoosefe94 formed a monophyletic lineage with fish ictchaphamaparvovirus genomes (GenBank no. NC055527 and ON596002).

### *Picornaviridae*

Picornaviruses comprise positive-sense single-stranded RNA viruses widely distributed in human and animal hosts (70). Some members cause severe diseases, such as paralysis induced by poliovirus and neurological disorders associated with enterovirus 71 (EV71) (71). This study focused on migratory birds during winter to uncover picornaviruses' diversity and ecological significance in avian hosts. In this study, a total of 41 picornaviruses were identified, including 12 from Siberian cranes and 18 from wild geese. A phylogenetic tree was constructed based on the RdRp proteins, and a BLASTx alignment was performed. Phylogenetic analysis revealed that PicornaCrane82V2, PicornaGoose95V2, and PicornaGoose94V3 clustered within the *Sapelovirus*, forming the same genera as simian sapelovirus and porcine sapelovirus. PicornaGoose91V2 and PicornaGoose96V2 shared over 98% nucleotide identity with ludopiviruses (GenBank no. NC040684) identified in *Anser albifrons* from Hungary, PicornaCrane74V1 and PicornaCrane66V1 exhibited over 90% amino acid identity with a gallivirus (GenBank no. MK204386) found in *Calidris ruficollis* from Melbourne, and Picornaviridae Goosefe87V1 showed 85.49% identity with a sicinivirus (GenBank no. MT345550) collected from chicken in North America. PicornaCrane66V3, PicornaCrane74V2, and PicornaCrane79 clustered within the *Grusopivirus*, sharing >97% identity with a strain from *Grus nigricollis* in Tibet (GenBank no. OR532954). Moreover, six picornaviruses clustered within the *Gruhelivirus* with identities <60%. Twelve picornaviruses belonging to *Hepatovirus* were also identified. These hepatoviruses were divided into three evolutionary clades, and almost all of them shared less than 50% amino acid sequence identity with their closest known hepatovirus relatives. Among them, six newly identified picornaviruses were classified within *Aalivirus* and *Avihepatovirus* on the phylogenetic tree, with amino acid

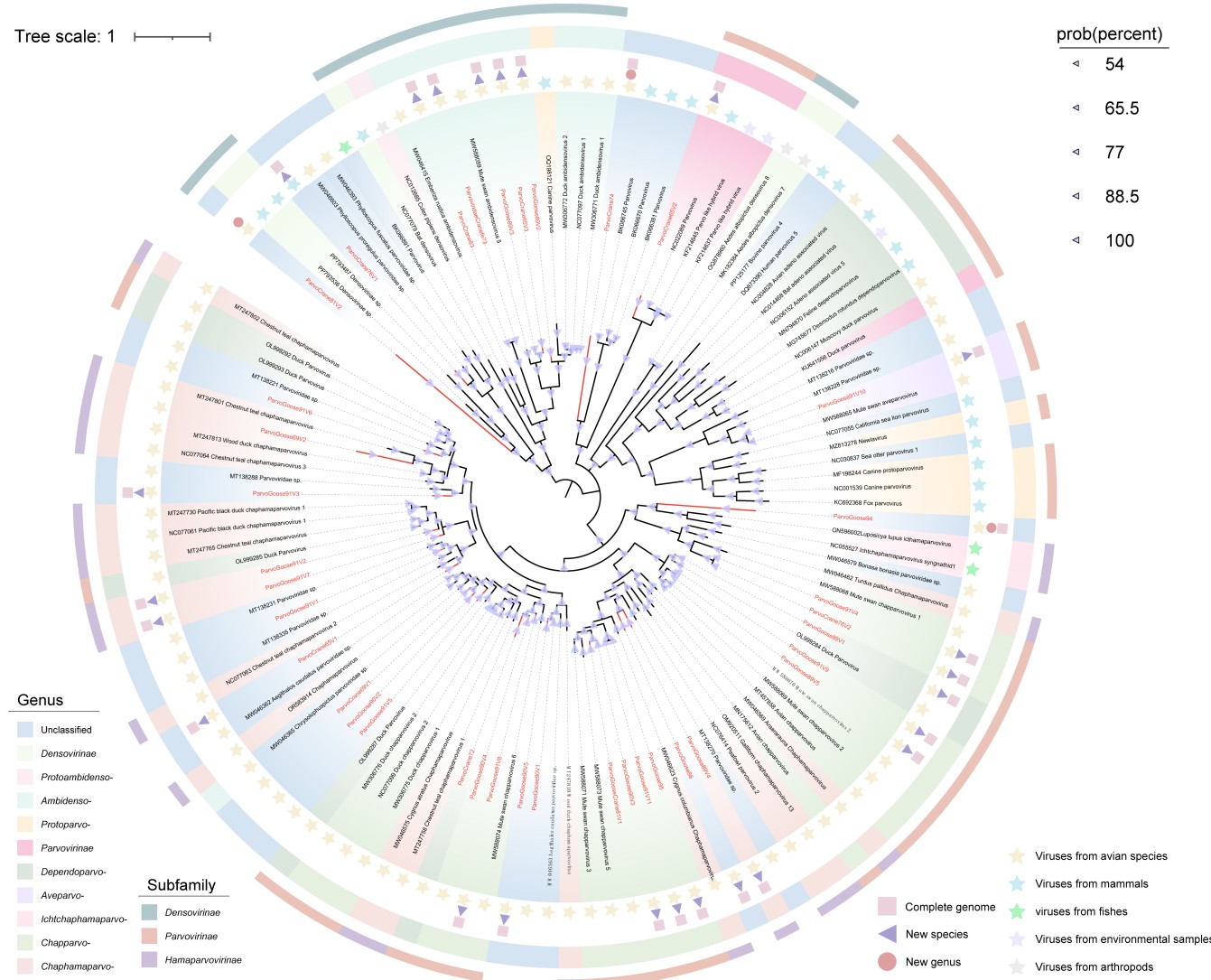

**FIG 8** Phylogenetic relationship of *Parvoviridae*. A Bayesian inference tree was constructed based on the amino acid sequence structure of non-structural protein 1 (NS1) of the family *Parvoviridae*. Sequences identified in this study are highlighted in red. Relevant annotations are provided in the accompanying legend.

sequence identities of less than 70% compared to their closest known picornaviruses. Notably, PicornaGoose87V2 clustered with known *Aquamavirus* strains, forming a distinct monophyletic clade (Fig. 9). Additionally, phylogenetic analysis revealed that Picorna-Crane73V4 represents a distinct and independently branching lineage, suggesting its evolutionary divergence from known relatives (Fig. 9).

## *Picobirnaviridae*

Picobirnaviruses are emerging double-stranded RNA viruses. Their genome consists of two segments: the L-segment usually encodes the RdRp protein, while the S-segment encodes the capsid protein (72). Picobirnaviruses have been associated with opportunistic gastroenteritis in humans and other animals, and studies have confirmed more frequent cross-species transmission of picobirnaviruses than any other RNA virus family (73). In this study, 11 RdRp-containing strains and eight capsid protein-encoding strains were identified from the feces of wild geese. BLASTx analysis in NCBI revealed that the RdRp sequences shared 58.57%–96.22% amino acid identity with their closest known Picobirnavirus (PBV) counterparts, with most sequences displaying over 70% identity.

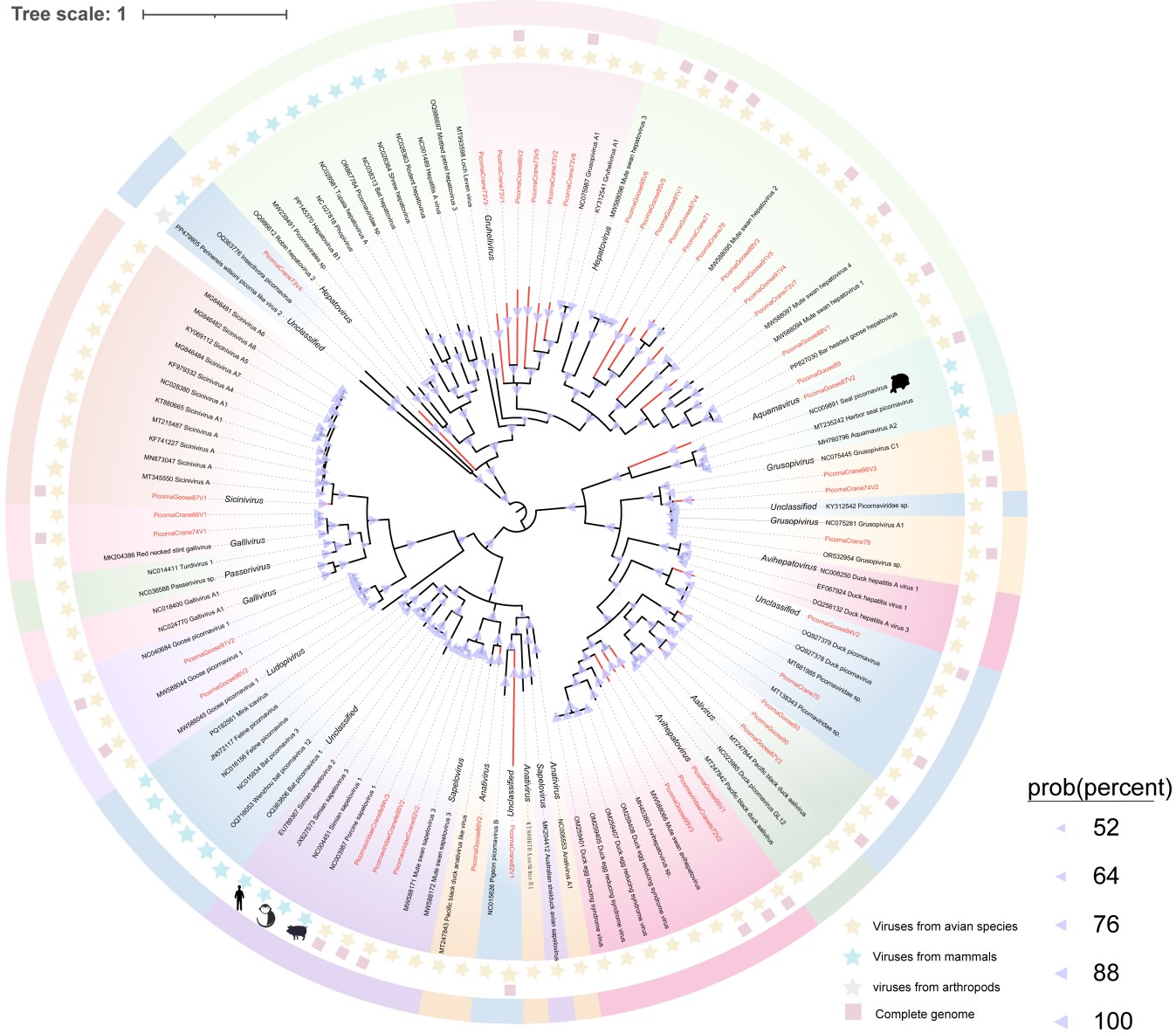

**FIG 9** Phylogenetic relationship of *Picornaviridae*. A Bayesian inference tree was constructed based on the RDRP proteins. Sequences identified in this study are highlighted in red. Relevant annotations are provided in the accompanying legend.

In contrast, capsid protein amino acid identities were below 50%, with the lowest being 35.85%. To analyze the phylogenetic relationships between the newly identified genomes and other known picobirnaviruses, a phylogenetic tree was constructed based on RdRp protein sequences (Fig. 10). The phylogenetic analysis reveals that picobirnaviruses from wild geese (e.g., PicobirnaGoose96V1 and PicobirnaGoose87V1) are evolutionarily closely related to picobirnaviral genomes found in various mammals, such as pigs, humans, and gorillas. For instance, PicobirnaGoose87V1 and PicobirnaGoose96V1 share over 93% amino acid sequence identity with pig picobirnaviruses. Additionally, PicornaGoose88V4 shares 80.94% and 81.73% amino acid sequence identity with picobirnaviruses from gorillas (GenBank no. OR532954) and humans (GenBank no. MH933806), respectively.

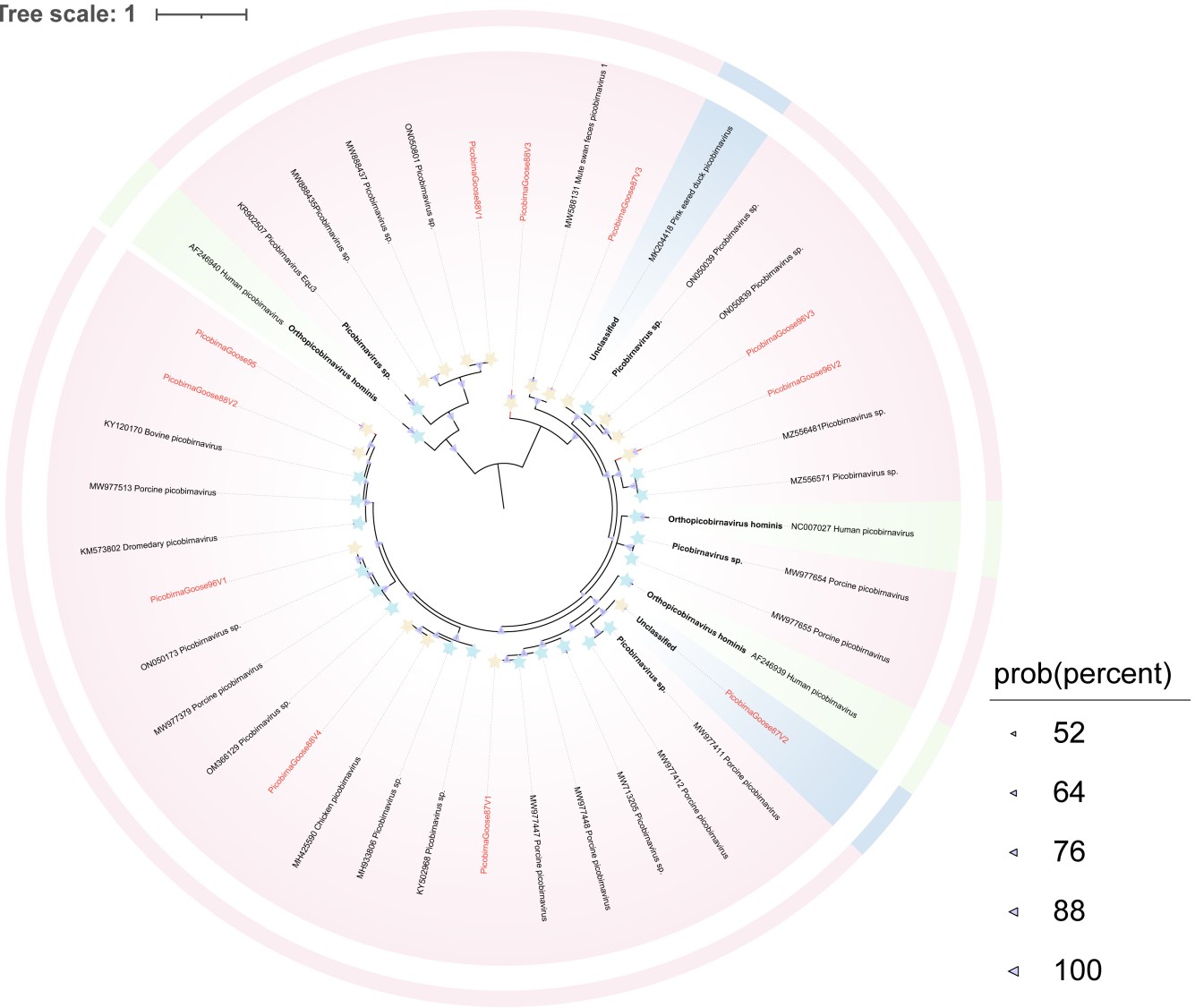

**FIG 10** Phylogenetic relationship of *Picobirnaviridae*. A Bayesian inference tree was constructed based on the RdRP proteins from *Picobirnaviridae*. Sequences identified in this study are highlighted in red. Relevant annotations are provided in the accompanying legend.

## CRESS-DNA viruses

Replication-associated (Rep) protein-encoding single-stranded DNA (CRESS-DNA) viruses are widely distributed and have been reported to infect nearly all branches of the eukaryotic tree of life (74). However, the genetic diversity of CRESS-DNA viruses remains poorly understood, with only a fraction characterized so far. In this study, we identified nine circoviruses, 11 genomoviruses, three smacoviruses, and 27 unclassified CRESS DNA viruses from fecal samples of Siberian cranes and geese (Fig. 11). BLASTx analysis revealed that, compared to known viruses, the Rep protein sequences of seven circoviruses showed amino acid sequence identities ranging from 36.28% to 59.70%, while two genomoviruses exhibited less than 60% identity. These findings suggest that these viruses may represent novel species. Notably, GenomoCrane76 shared 99.54% amino acid sequence identity in its Rep gene with a murine genomovirus (GenBank no. OK491636), suggesting a close evolutionary relationship between the Siberian crane and murine genomoviruses. Phylogenetic analysis further revealed a broad host range among different CRESS-DNA viruses. Genomoviruses and circoviruses were found to

infect a wide array of hosts, including birds, reptiles, protists, plants, mammals, and arthropods, whereas smacoviruses were primarily associated with birds and mammals. Additionally, we identified several unclassified CRESS-DNA viruses that could not be assigned to any established viral family.

## DISCUSSION

Migratory birds are increasingly recognized as reservoirs of a wide array of largely uncharacterized viruses with unknown host ranges, ecologies, and pathogenic potential (75). Previous research has demonstrated that wild birds can disseminate viruses like IBV and avian astroviruses (51, 76). Furthermore, significant outbreaks of highly virulent avian influenza have underscored the role of migratory birds in the long-range spread of viruses (77). Nonetheless, these documented instances may constitute but a minor segment of the extensive variety of viroids harbored by these species. Recent advancements in metagenomic sequencing have uncovered a wide viral diversity within migratory bird populations that remains largely underexplored, partly due to the ecological complexity in which these viruses circulate (78). Therefore, a deeper understanding of the ecological contexts that sustain and facilitate virus persistence and transmission is essential. Poyang Lake's complex ecological network and high density of migratory birds' aggregations provide conditions for virus transmission and recombination between hosts (79). Consequently, ongoing virologic surveillance of ecologically significant migratory bird populations and habitats is crucial for enhancing our comprehension of viral diversity, evolution, and potential ecological ramifications.

This study aimed to investigate the virome composition in the intestines of wintering Siberian cranes and wild geese at Poyang Lake, providing insights into viral diversity and ecology in migratory birds. A total of 320 samples were analyzed, including 208 fecal samples from Siberian cranes and 112 fecal samples from wild geese. The findings are significant, as they identified 183 viral genome sequences associated with known and putative vertebrate-infecting viruses.

In this study, a novel Gammacoronavirus was identified in fecal samples from Siberian cranes, providing critical insights into the adaptive evolution of avian coronaviruses. Phylogenetic clustering of CoronaCrane85 with a mute swan gammacoronavirus (Fig. 6) was strongly supported by shared structural hallmarks in the S protein, including co-occurring insertions (residues 341–343 and 441–443), deletions (346–352 and 458–462), and conserved RBD mutations (Y472, G419), suggesting potential recent host divergence or cross-species transmission events (Fig. 6). Notably, Gammacorona1792 harbors a unique V437I substitution within a predicted receptor-interaction loop (Fig. S3). Although valine and isoleucine are biochemically similar, this substitution may induce subtle changes in local conformation or electrostatic properties, potentially affecting receptor binding. The V437I substitution, while not reported in other coronaviruses, has structurally comparable alterations in the receptor-binding domain (RBD), such as the N501Y mutation in SARS-CoV-2 and deletions in the S1 subunit of avian IBV, which have been shown to affect receptor binding, host adaptation, and immune evasion (80, 81). These examples suggest that RBD remodeling, even through minor amino acid substitutions or indels, may contribute to host-specific adaptation or immune escape. The evolving nature of the coronavirus spike protein and the ecological convergence of migratory birds at Poyang Lake jointly influence viral diversity. Deciphering host-mediated adaptation processes necessitates prolonged genomic and ecological surveillance.

Our study reveals that many newly discovered viruses exhibit low genomic similarity to known viruses and form independent branches in the phylogenetic tree, suggesting that these viruses may represent previously undescribed viral lineages. This study identified 11 nearly complete anellovirus genomes, with six newly identified anelloviruses clustering into a new branch in the phylogenetic tree, potentially representing a new genus (Fig. 4). Studies suggest that anelloviruses may contribute to host-virus tolerance through immune evasion, particularly in long-term infections (82). Anelloviruses are genetically diverse viruses. The high variability of these viruses

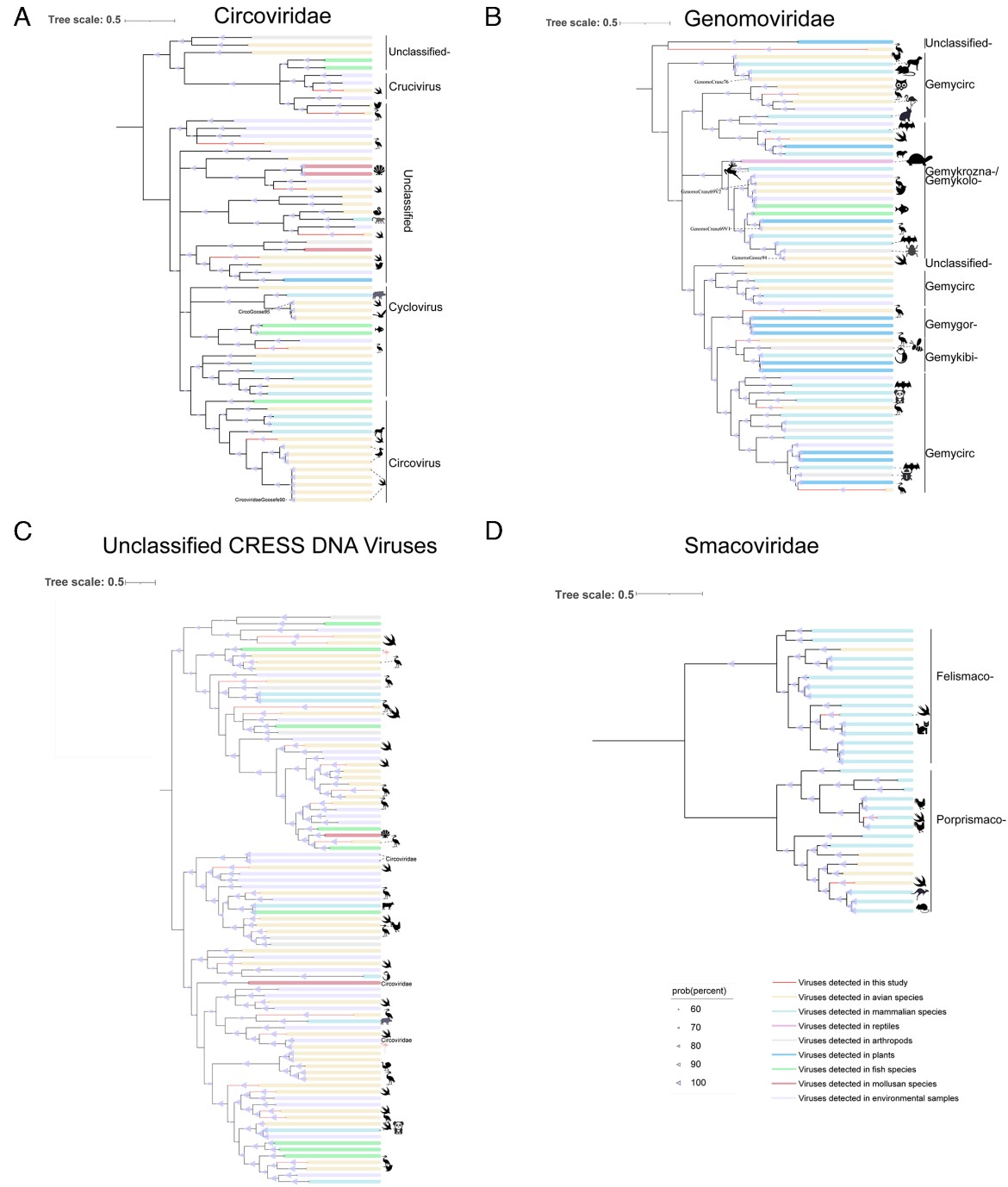

**FIG 11** Phylogenetic relationships among CRESS-DNA viruses. Maximum likelihood trees were inferred based on the Rep proteins from four viral groups: *Circoviridae* (A), *Genomoviridae* (B), unclassified CRESS-DNA viruses (C), and *Smacoviridae* (D). Different-colored lines indicate the corresponding hosts, with further details provided in the legend located at the bottom right.

and their potential for genetic recombination across different species further increases their diversity and could lead to the emergence of novel variants (83). Specifically, the correlation of the AnelloCrane74 and bat gyrovirus genomes in the evolutionary tree suggests that anelloviruses may have undergone cross-species transmission. The potential chain of virus transmission between birds, bats, and humans cannot be ignored, especially given the migratory behavior of birds and the role of bats as known virus hosts, which may act as vectors for each other, either through the environment or

by direct contact (84). Previously, our team used viral metagenomic and high-throughput strategies to reveal the unexpected diversity of parvovirus viral dark matter in the avian gut (85). In the current study, we identified 33 new virus species and three potential new genera, further expanding the diversity of parvoviruses in birds. Our findings suggest that there may be a significant number of undescribed parvovirus lineages within bird populations, providing new insights for future viral research, particularly in the study of cross-species transmission and host range expansion. Given their long-range migratory behavior, birds have the potential to disseminate parvoviruses across borders. In addition, 19 picornaviruses were identified, all with less than 70% amino acid sequence similarity to known picornaviruses. In particular, phylogenetic analysis revealed that PicornaGoose87V2 clustered within the genus *Aquamavirus*, which currently contains a single species (*Aquamavirus A*) found in seals (Fig. 8). This phylogenetic placement suggests that aquamaviruses may have a broader host spectrum than previously recognized, potentially including birds. While further genomic and biological characterization is needed, this observation provides new insights into the ecological dynamics and evolutionary history of the *Aquamavirus* lineage. This research also uncovered many CRESS-DNA viruses, the majority of which show low similarity to known viruses. It facilitates the classification of unclassified viruses and may result in the discovery of new viral families, genera, or species. The unclassified CRESS-DNA viruses had specific host preferences, underscoring the unique virome ecology of the gastrointestinal tracts of Siberian cranes and geese inhabiting Poyang Lake in winter. These findings underscore the multitude of previously unidentified viruses in migratory birds, accentuating the untapped viral diversity and the imperative for additional research into their taxonomy and evolution.

Our study identified several novel viruses exhibiting high genomic similarity to previously reported avian viruses. For instance, one astrovirus (AstroCrane81) and three picornaviruses (PicornaCrane82V2, PicornaGoose95V2, and PicornaGoose94V3) formed separate monophyletic lineages with viruses previously identified in mute swans from Dorset, UK. Notably, these picornaviruses were classified into the same genus as several mammalian picornaviruses (e.g., those infecting humans, monkeys, and pigs), which may indicate a shared evolutionary origin. Their actual host range and potential for cross-species transmission remain to be determined through experimental studies (Fig. 9). In addition, several picornaviruses shared 85.49%–98% genomic similarity with strains identified in greylag geese from Hungary, red-necked stints from Melbourne, chickens from North America, and bar-headed geese from Tibet. Taken together with their phylogenetic proximity and broad host distribution, such findings support the hypothesis that these picornaviruses may follow migratory bird flyways, facilitating wide geographic dispersal and possible interspecies transmission among avian hosts. Similarly, caliciviruses displayed high amino acid similarity to viruses previously found in chickens, ducks, and swans while forming a distinct phylogenetic branch among unclassified caliciviruses (Fig. 7). This distinct phylogenetic clustering may reflect a previously underexplored lineage of caliciviruses circulating among overwintering migratory waterfowl at Poyang Lake. Although the zoonotic potential of these viruses remains unclear, their genetic diversity and distribution underscore the ecological role of migratory birds as reservoirs and dispersal agents of diverse RNA viruses. Further research is required to evaluate their host range, evolutionary dynamics, and possible relevance to animal or public health. Several picobirnaviruses identified in wild geese exhibited notable amino acid similarity to strains previously detected in mammals, including pigs, gorillas, and humans. Given their reported association with gastrointestinal diseases, such as diarrhea in humans, future research should focus on characterizing their protein structures and functional properties to assess their human infectivity and elucidate their pathogenic mechanisms (86). The extensive migratory behavior of birds, involving broad geographic movement and exposure to diverse ecological niches, likely facilitates the dissemination and genetic diversification of picobirnaviruses.

Although AIVs were emphasized in this study due to their recognized importance in wild bird health and zoonotic risk, no complete AIV genomes were recovered from our study. This absence likely reflects methodological and ecological factors rather than a true absence in the host populations. AIVs possess segmented negative-sense RNA genomes that are prone to degradation and uneven representation in fecal samples, complicating full-genome assembly in untargeted metagenomic workflows. Moreover, the lack of virus-specific enrichment and host RNA depletion steps limits sensitivity for low-abundance RNA viruses. The timing of sampling may also have contributed: fecal shedding of AIVs in migratory birds typically peaks during spring migration, whereas samples in this study were collected in winter, when viral replication may be suppressed due to physiological adaptation for overwintering. Previous studies have reported variable AIV detection rates in wild bird feces, influenced by host species, geographic region, season, and sampling strategies (87, 88). Therefore, the absence of AIVs in this study does not preclude their circulation but highlights current limitations in detection. Future investigations incorporating targeted amplification, viral enrichment, and deep sequencing may improve sensitivity. This study also focused on a limited number of species and a single overwintering location. Viral diversity in Siberian cranes was significantly higher than in wild geese (Shannon index, $P = 0.022$), with distinct community structures (PCoA, R = 0.3). Siberian cranes are currently listed as critically endangered by the International Union for Conservation of Nature (IUCN25, $P = 0.001$), potentially shaped by host ecology and habitat interactions. Expanding virome surveillance across seasons, geographic regions, and host taxa, particularly for endangered species such as the Siberian crane, will be essential for identifying ecological drivers of viral diversity.

## ACKNOWLEDGMENTS

This work was supported by the National Key Research and Development Program of China no. 2023YFD1801302 and no. 2024YFF1307203.

B.N., W.Z., and W.H. designed the study and methods. J.G., Y.C., and S.H. constructed the libraries. J.G., Y.X., and X.J. completed the data analysis. The paper's first draft was prepared by J.G. and substantially reviewed and revised by all authors.

The authors declare no competing interests.

## AUTHOR AFFILIATIONS

[1]Department of Microbiology, School of Medicine, Jiangsu University, Zhenjiang, Jiangsu, China
[2]Jiangxi Academy of Forestry, Nanchang, Jiangxi, China
[3]Department of Clinical Laboratory, Wuxi Blood Center, Wuxi, China

## AUTHOR ORCIDs

Jing Gao  http://orcid.org/0009-0009-1930-4462
Bin Ni  http://orcid.org/0000-0003-3975-0602

## FUNDING

| Funder | Grant(s) | Author(s) |
| --- | --- | --- |
| National Key Research and Development Program of China | 2023YFD1801302 | Bin Ni |
| National Key Research and Development Program of China | 2024YFF1307203 | Weijie Han |

## AUTHOR CONTRIBUTIONS

Jing Gao, Validation, Visualization, Writing – original draft, Writing – review and editing.

## DATA AVAILABILITY

The library data involved in this study have been deposited into the National Genomics Data Center (NGDC) of China and the Short Read Archive (SRA) of the GenBank database under the BioProject accession no. PRJCA037117/PRJNA1248161.

## ADDITIONAL FILES

The following material is available online.

### Supplemental Material

**FIG. S1 (mSystems00756-25-s0001.tif).** Pairwise comparison of amino acid sequences of anelloviruses identified in this study with the representative strains of gyroviruses.
**FIG. S2 (mSystems00756-25-s0002.tif).** Genomic organization of the coronavirus.
**FIG. S3 (mSystems00756-25-s0003.tif).** Analysis of the RBD of the spike protein of CoronaCrane85.
**FIG. S4 (mSystems00756-25-s0004.tif).** Phylogenetic relationship of Coronaviridae.
**FIG. S5 (mSystems00756-25-s0005.tif).** Phylogenetic relationships among CRESS-DNA viruses.
**Table S1 (mSystems00756-25-s0006.csv).** The *P*-dist of the four newly identified astroviriuses.

### Open Peer Review

**PEER REVIEW HISTORY (review-history.pdf).** An accounting of the reviewer comments and feedback.

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
