## [Reviewer comments · mSystems]

Unexplored Viral Diversity in Siberian Cranes and Wild Geese: Metagenomic Insights from a Global Wintering Haven

Jing Gao, Wei Han, Xiaojie Jiang, Yuan Xi, Yue Chen, Shi Huang, Xiao Huang, Yang Zhang, Tianxiang Zhang, Man Zhang, Wen Zhang, and Bin Ni

Corresponding Author(s): Bin Ni, Jiangsu University

Review Timeline:

Submission Date:

May 30, 2025

Accepted:

July 30, 2025

Editor: Kirsten Hofmockel

Reviewer(s): Disclosure of reviewer identity is with reference to reviewer comments included in decision letter(s). The following individuals involved in review of your submission have agreed to reveal their identity: Walter Harrington (Reviewer #2)

Transaction Report:

DOI: <https://doi.org/10.1128/msystems.00756-25>

Re: mSystems00756-25 (Unexplored Viral Diversity in Siberian Cranes and Wild Geese: Metagenomic Insights from a Global Wintering Haven)

Dear Dr. Bin Ni:

Your manuscript has been accepted, and I am forwarding it to the ASM production staff for publication. Your paper will first be checked to make sure all elements meet the technical requirements. ASM staff will contact you if anything needs to be revised before copyediting and production can begin. Otherwise, you will be notified when your proofs are ready to be viewed.

Sincerely,
Kirsten Hofmockel
Editor
mSystems

Reviewer #1 (Comments for the Author):

The authors have adequately addressed all my questions.

Reviewer #2 (Comments for the Author):

See the attached file.

Unexplored Viral Diversity in Siberian Cranes and Wild Geese: Metagenomic Insights from a Global Wintering Haven

Overall Comments

In this study, Gao and co-authors have done an important investigation into the viral diversity discovered in the feces of Siberian cranes and geese. Siberian cranes are an under sampled population, and a strength of this study is that 98% of Siberian cranes winter on the lake in which the study was conducted. The novelty of this study lies in the discovery of previously unknown viruses both in these bird populations and in any animal populations. The authors have done an extensive revision addressing my comments. Below are a few specific comments on the revision.

Specific Comments

- Thank you for adding the context to explain why AIV might not be found in your dataset. I think your explanation of the untargeted sequencing is sufficient, without the mention of timing. In North America, we consistently find LPAI even in overwintering populations. Unless the ecology of AIV is very different in migratory birds in Asian flyways, I would expect AIV to be present in overwintering populations as well. However, the untargeted approach and the complications from a segmented genome in metagenomics studies I think are sufficient explanations for the lack of full AIV genomes in this dataset. I do think it might be helpful to mention if you found any fragments of the AIV genome in your dataset in the discussion, but I would not be a stubborn on this issue and will leave it up to your discretion.
- Thank you for toning down the descriptions of the results, I believe this makes it more objective and understandable.
- The results section has been sufficiently updated to reflect results and the corresponding discussion points have been moved to the discussion.
- Thank you for defining the threshold and objectively stating the percent identities for “closely related” species.
- I think the unified graphics are much easier to understand.

Minor Comments

- All minor comments have been sufficiently addressed.